# Tracking Websites’ Digital Communication Strategies in Latin American Hospitals During the COVID-19 Pandemic

**DOI:** 10.3390/ijerph17239145

**Published:** 2020-12-07

**Authors:** Santiago Tejedor, Ana Pérez-Escoda, Augusto Ventín, Fernanda Tusa, Fátima Martínez

**Affiliations:** 1Department of Journalism and Communication Sciences, Autonomous University of Barcelona, 08001 Barcelona, Spain; Santiago.Tejedor@uab.cat; 2Department of Communication, Antonio de Nebrija University, 28015 Madrid, Spain; 3Faculty of Communication, Universidad de La Sabana, Chia 53753, Colombia; 4Faculty of Social Sciences, Technical University of Machala, Machala 070201, Ecuador; ftusa@utmachala.edu.ec; 5Department of Journalism and Public Opinion, University of El Rosario, Bogotá 110111, Colombia; fatimamargu@gmail.com

**Keywords:** digital communication, Internet, hospital, web design, COVID-19, infodemic

## Abstract

Since the advent of the Internet, websites have become the nerve center of the digital ecosystems of media, companies and all kinds of institutions. Currently, the impact of the global coronavirus pandemic has placed healthcare issues at the center of social debate, including hospitals and their websites as digital sources of trustworthy information. COVID-19 has intensified the need for quality information and the legitimacy of sources fighting the *infodemic* situation. In this regard hospitals become essential social actors in the spread of healthcare information. Within this framework, a qualitative study is presented with descriptive components and based on content analysis. This study examines 58 websites from the best hospitals included in the “América Economía” ranking health sector from Latin America. The study applies an analysis methodology based on previous research focusing on specialized web studies, defining an analysis model on six variables and 65 thematic indicators. The research concludes that hospitals occupying first positions in the medical services directory are not necessarily those that have the best websites. Similarly, it is worth noting that a quarter of the studied sample do not devote a specific space to reporting coronavirus information. Brazil, Colombia and Chile are the countries with the highest number of hospitals among those with the best websites. In conclusion, digital media, specifically websites, could constitute legitime resources of healthcare information consumption, so their accuracy and proper development seem to be significant to become genuine sources that not only could provide better healthcare services but help avoid the spread of misinformation about the COVID-19 pandemic.

## 1. Introduction

COVID-19 has had a global impact on different social areas of our daily lives, such as mobility, leisure, entertainment, education [1], communication [2], economics, market, and, especially, the health sector. Hospitals have been overwhelmed by the emergence of coronavirus disease cases. Access to clear, structured and accurate information has become a decisive resource for citizens and a crucial challenge for health centers to ensure some kind of social order in a situation of lockdown [3] and *infodemic*.

Websites have become crucial platforms not only for digital media ecosystems, but for companies and public or private institutions [4]. The proliferation of corporate websites has taken place in a media context nurtured by a process of permanent mediatization [5], which has been altered in recent years by several problems such as fake news, clickbait, post-truth and infoxication [6]. Moreover, the necessity to deploy media literacy processes that allow digital illiterates to be transformed into critical and demanding users of the platforms and their contents has been added to this changing social framework [7,8]. This challenge demands a work of revision and improvement in the designing and structuring of the websites that offer basic services, such as administration, justice or health, as well as special attention in contents that are disseminated through them [9,10].

All along these transformations have occurred in a scenario shaped by the correlative transition from Web 1.0, to Web 2.0 or Social Web, and lately from the Web 3.0, the so called Semantic Web to Web 4.0, the Internet of Things, evolving in a new communicative framework [11,12]. In this arena, through websites, weblogs and other types of platforms, institutions, companies and other entities have developed their corporate pages on the Internet. Since 2007, health institutions have created their own platforms, with websites being the main point of their respective digital ecosystems [13,14]. Cyberspace’s communication possibilities have allowed hospitals to provide a point of information and interaction which, however, have not always made the most of the potential offered by the Internet in the creation, management and distribution of different types of digital content [15]. The impact that the coronavirus pandemic has socially generated, collapsing the daily life of many hospitals in the world due the enormous strain for COVID-19 patients is added to this particular challenge. In this context, the websites of health institutions must be able to play an informative, interactive and awareness-raising role that helps citizens to become informed, resolve doubts and manage their medical or health needs.

The *infodemic* resulting from the COVID-19 crisis has posed new challenges to communication in the field of science, health and medicine, in particular. The concept of *infodemic* has been used to allude to the overabundance of information, rapidly spread and unreliable, which has been one of the characteristics of the global crisis introduced by the coronavirus. The World Health Organization (WHO) has used this concept to warn citizens of an excess of information, within which a large number of hoaxes or rumors circulate. In this scenario, the communication strategy of health centers, focusing on hospitals, opens up an auspicious field of study focusing on the relevance of this sites to fight against the *infodemic* [16].

Having in mind previous paragraphs and pioneering studies such as that of Calvo-Calvo [17] and Jha et al. [15], which have investigated the quality and characteristics of websites in several countries health system, this research analyses the websites of the 58 most important hospitals in Latin America. From a descriptive, cross-sectional perspective, which identifies, and studies elements related to the content and access, usability and visibility of these websites, this is an innovative study on how hospitals have communicated the pandemic crisis and how they have managed information related. Main conclusion point to consider hospital’s websites as resources offering essential medical support to the public during the COVID-19 outbreak, that can help to reduce the *infodemic* effects, including social panic. These pages could also promote social distancing, good practices and enhance the public’s ability of self-protection, becoming primary sources of information and avoiding misinformation.

## 2. State of Art

The World Summit on the Information Society benchmarks, held in 2003 in Geneva, Switzerland [18], provided the support for the World Health Organization (WHO) and the Pan American Health Organization (PAHO) to use the concept of e-health in allusion to the potentialities derived from the application of educational and communication technology to the health field. This concept emphasizes the usefulness of ICTs to strengthen healthcare services, health monitoring and documentation, and equally education, knowledge and research in the health area. The concept of *e-health* gave way to the concept of *m-health* which transferred these opportunities to the field of mobile devices. In this scenario, although in 2011 68% of the 19 countries analyzed in the Pan American Health Organization (PAHO) survey placed e-health as a priority in their national agendas, nowadays it still remains a field of study to be developed [15,19].

Information technologies have provided a framework plenty of information, communication and training possibilities in the healthcare field [20]. These are resources that could improve the services of medical professionals and centers, but which could also help citizens to become more prepared patients managing their own healthcare processes. Some studies, such as the Wald et. al. [14], have pointed out that the use of the Internet in the health field constitutes a great tool to help patients make more informed decisions. Furthermore, digital tools and platforms are pretty valuable in promoting patients’ own motivation [9], which demands intergenerational work processes. These reflections are supported by previous literature as shown in the extensive literature review published by Kampmeijer et al. [21] in which authors evidenced how e-health and m-health tools (understood as namely apps, websites, digital content, video consults and webinars) truly improve healthcare quality for adults in the way they receive more supportive assistance and are better informed. Moreover, since COVID-19 lockdown situation occurred the role of institutional websites from hospitals, has become an essential issue, in most cases the unique communication channel between patients and healthcare services [22,23].

In this regard it could be found several studies on the evaluation and analysis of websites as specialized subjects from the wider concept of e-health defining the communicative particularities of this issue. In this sense, authors such as Codina [24,25], Rodríguez-Martínez [26], Túñez [27], Marín Dueñas [28], Nevado [29], Cobos and Recoder [30] or Tejedor, Cervi and Gordon [31], among others, have elaborated system indicators for systematic analysis of web pages, that could be applied to sanitary environments.

At the beginning of the century hospitals began to develop their own web pages as a key strategy in their online visibility, authors such as Mariscal [32] have insisted on the need for these health services to strengthen their presence on the web with the aim of give patients more and better qualified services, reducing costs and boosting their digital branding. Digital media and connected devices have evolved to a ubiquitous part of our daily life and the Internet have paved the way to a wider interactive communication in the relation with citizens [2]. Healthcare services delivery and information availability supported by digital technologies such as hospital´s websites are key issues not only in the enhancement of patient satisfaction but in the efficiency of e-Health systems [20,21,22]. As stated from Kotsiliers et al. [33] if these means are properly designed, they offer a double innovative pathway: from one hand giving patients the power not only to educate themselves but take part in the decision-making process of their health; and in the other hand these new channels for communication and interaction make healthcare stakeholders gain in essential aspects such as cutting costs, gaining efficiency and improving the healthcare system. At this point, is important noting that Calvo-Calvo [17] and Gámez de la Hoz [34] highlight this in their research into the quality and characteristics of the health services that hospitals and other types of clinical establishments offer on the internet. However, for the moment, this type of study does not exist in the Latin American context.

The digital divide has been one of the main obstacles as highlighted by ITU [32]. However, since 2014, studies have identified an increase on the Internet access in the Latin American scenario. “Telefonica Global Millennial Survey” report [35] pointed out that 78% of young—less affected population by COVID-19– Latin Americans between 18 and 30 years old have a smartphone. According to the study, these devices were the most used by the youngest, and in this order: accessing social networks, reading messages, and making calls. In 2015, Latin America achieved one of the highest rates in social networks usage in the world: 78.4% of Latin American internet users participated in social networks, exceeding the percentage from North America (64.6%) and Europe (54.5%) [36].

On this line the study “State of Broadband in Latin America and the Caribbean 2017” highlights that regarding the number of households connected to the Internet in the region the percentage increase was 103% between 2010 and 2016 [37]. Brazil was the Latin American country with the largest number of Internet users. About 105.4 million Brazilians used the internet in 2019 (see Figure 1). Mexico is the second country in the region with the greatest number of users, reaching 89 million. At the end of January 2020, most Latin American countries had more than 60% of internet users. The Caribbean region registered some of the highest internet penetration rates in Latin America and the Caribbean. Chile was the only country exceeding 80% of the population using the Internet [38]. However, the region still has significant digital divides affecting not only the need to improve digital and media literacy processes, but to improve digital ecosystems and contents [32,37].

## 3. Materials and Methods 

The research presented, which is descriptive, explanatory and exploratory, has been conceived as a diagnostic and, at the same time, proactive analysis [39]. The study is based on content analysis understood as a research technique that allows the construction of categories and indicators to describe the values or messages of a content, platform or process [40]. Having in mind the arguments described in the state of the art, the following research questions have arisen:(1)Which are the defining features of Latin American hospital websites in terms of content and access to information, visibility and usability?(2)What structural or content strengths and weaknesses are identified in these websites?(3)What interactive resources have a leading role?(4)Which kind of connection exists with other types of platforms, especially with social networks?

To answer these research questions, the study has pursued the following methodological approach:(1)Review of the specific literature of authors with prestige in the evaluation of websites and cybermedia.(2)Analysis of the selected websites and their connections with other platforms.(3)Adaptation of the SAAMD pattern (initials from “Sistema Articulado de Análisis de Medios Digitales” this is Articulated Digital Media Analysis System), proposed by Codina and Pedraza Jiménez [41] to the thematic area of hospital websites.(4)Definition and design of an own analysis table based on a specialized bibliographic review and considering the particularities of the sector. To this end, the study has defined 6 thematic areas to be evaluated on each website and 65 indicators have been specified for the development of the analysis. The study analyzes the appearance or not of COVID-19, taking into consideration the presence of information related to the pandemic on the websites analyzed.(5)Application of the web analysis pattern to each of the selected pages.(6)Extraction of results to answer the questions raised in the research.

According to Codina [24,25], Tejedor [42], Calvo-Calvo [17] and Cobos and Recoder [30], the research model has been designed in six research variables of study (see Table 1) defined and analyzed in different indicators for the study each one, including a whole of 65 items that could measure the main research variables (see Table 2).

The study has adopted a list of closed indicators that denote absence or presence in the application to the case study of the sample in order to reduce the subjectivity of the analysis. For this reason, the score for the measurement is binary (0/1), being 0 “no” and 1 for “yes”. The following analysis table has been created from the 6 variables defined and applied in individual worksheets to obtain all data:

The sample invited to the study, conceived as the set of elements of the population asked to participate in the research [39], corresponds to the 58 hospitals that occupy the top positions in the main rankings of the sector. In this sense, the study has been based on the ranking prepared by the “América Economía” portal, taking into consideration any Latin American hospital or clinic of high complexity which offers services in a wide range of Medical Specialties and which has been identified as a reference by the Ministries of Health of Argentina, Brazil, Colombia, Costa Rica, Chile, Cuba, Ecuador, Mexico, Panama, Peru, Uruguay and Venezuela, or other relevant sources. The ranking measures seven indicators (patient health and dignity, human capital, capacity, knowledge management, efficiency, prestige, dignity and patient experience). Of all the Latin American hospitals, only those that manage to obtain a score of more than 50 points in the final indicator are included in the ranking. Table 3 shows the list of hospitals which make up the ranking in the Latin American region and which have been studied in this research.

## 4. Results

The results derived from the study, which have been organized and presented by the 6 main variables of study (usability, interactivity, information, typology, quality and accessibility), identify a range of aspects of great value in the analysis of the websites of the main hospitals in Latin America as can be seen in paragraphs below.

### 4.1. Usability

In this sense, with regard to “Usability”, it can be said that this is an aspect present in the design of websites. As can be seen in Figure 2, all the hospitals analysed (100%) have internal links that connect their different sections and sections. In this way, a clear navigation route is guaranteed between the contents of the different sections of the portal and the interrelationship of themes, messages and other types of services.

Particularly striking is the high score of the links, 94.64% of the sample which apply a wording adapted to screen reading, allowing Internet users to know the type of website they will be accessing. In addition to this, in both cases with a percentage of 91%, the speed of downloading the contents of the websites, on the one hand, and on the other, the feature that keeps the navigation menu open while the user navigates the page. The presence of the search engine tool (with a percentage of 82.14) is also high, as well as the use of external hyperlinks with a 75% of the sample that connect to other pages and allow for more information. The most critical aspects refer to the scarce number of websites only the half of the sample (50%) that offer a multilingual platform and, on the other hand, only 17.87% of the hospitals studied present a site map which is essential for navigation. The most worrying aspect is that 5.35% of the hospitals analyzed provide information regarding the size, format or download time of the documents they offer on their websites.

### 4.2. Interactivity and Relationship with Users

Regarding to the second study variable, “Interactivity and interactions” with the visitors to their pages, the study shows (as shown in Figure 3) that practically all the hospitals studied provide basic contact data (including telephone, fax and postal address). The figure is slightly lower in relation to the presence of the specific contact data of the different areas or services. However, in both cases, it is possible to state that the websites provide complete information on this basic information. A total of 82.14% of the hospitals studied offer the possibility of making medical appointments through their respective pages and, in this same line, a large number, 76.78% present a general space for consultation through an e-mail or a basic information form. This same service is also offered in 69.64% of the sample with the specific e-mails of each unit, specialist or subarea. The weakest point refers to the existence of a suggestion box which is only present in less than half, 44.64% of the selected hospitals. 

### 4.3. Information Offered and Typology

With regard to the information provided on the websites, related to the third variable studied, a 100% of hospitals offer basic information about the center and 83.92% also offer information about their specific services as shown in Figure 4. The large number of cases is noteworthy. A percentage of 82.14 of centers analyzed, they include a welcome message, a letter of introduction or a message focused on transmitting the mission and vision of the center. In this case, it is usually the manager or the general director of the hospital. Of the total, 78.57% of the hospitals offer data on the general activity of the center in an annual report or summary report; while 75% rely on content in the form of a map to describe and convey the characteristics and services of the center. 

The number of medical centers that identify their staff by title and name is 69.64%; while only 48.21% of the sites analyzed provide information on the organization chart of management positions and their functions in the hospital. The study identifies that there is no habit of providing information with data and percentages relating to the number of care operations, interventions or surgeries carried out. There are, however, some exceptions such as the “Clínica Universidad de la Sabana” (Colombia) which does offer infographics and content related to the so-called data content or data journalism as can be seen in Figure 5.

The information on the websites analyzed, shows that 96.42% of the hospitals offer a general information portfolio and 92.85 percent present it with their specific services. It is noteworthy, among other aspects, that in 85.71% of the cases there is a news section related to the hospital (as shown in Figure 6). This aspect is very striking in the current scenario impacted by the growth of fake news and, where the World Health Organization has recovered the concept of *infodemic* alluding to the rapid growth of manipulated and falsified content circulating, especially, in cyberspace.

It could be observed that 78.57% from the hospitals studied that content is provided relating to the rights and duties of users. One of the most important topics at present is the presence of content relating to the coronavirus pandemic. In this sense, the 73.21% of the hospitals offer specific sections or information dedicated to the COVID-19 pandemic. It is, however, worrying that, a quarter of these centers, important noting that the best in ranking, do not offer spaces or content focused on the pandemic (characteristics, advice or recommendations in the event of contagion). Having in mind the situation has provoked many collapses in several hospitals, the information, regarding to pandemic (prevention, actions, tests, disease characteristics, etc.) becomes essential and invites a deep reflection from the communication perspective. Focusing on other indicators, results show that 71.42% of the hospitals offer general health advice or recommendations, while 66% have spaces dedicated to research or training in health-centered issues in different areas. Finally, it is striking that only 44.64% of the hospitals analyzed detail how to proceed with the management of a complaint or suggestion. Of the total, only 32.14 percent include in their web sites information about the cafeteria, restaurant or shop services of their centers, an aspect which shows how the websites leave the relatives and/or companions of the patients in a second place of importance within their digital communication strategy. The fact that only 5.35% of the hospitals offer information on waiting lists invites a reflection on transparency on a subject of great seriousness and concern among citizens.

According to the presence of informative content, the “San Vicente” Hospital in Medellin (Colombia) presents its own webzine called “The Pulse” (Figure 7), which offers content on health and about the hospital in a free access publication available to Internet users and completing other information that appears on the website.

### 4.4. Social Media and Updating of Contents

Low prominence of contents related to the most current trends (as seen in Figure 8) is one of the most significant results of the study. In this sense, the presence of gamified contents, augmented reality contents, contents with immersive video or photographic or video 360° contents or podcasts are practically non-existent. In general, less than a dozen of the hospitals analyzed present some content of this type. In addition, chats and forums have lost their importance as spaces for dialogue and only appear in four of the centers analyzed. The bet of contents based on the playful logic of games appears in 32.14% of the websites; while augmented reality (7.14%), podcast (10.71%) and immersive photographs and videos (12.5%) is equally scarce. Furthermore, only 8.9% of the hospitals analyzed show a transmedia strategy, which is related to information offered in different digital platforms, designed within a wider digital ecosystem.

In this regard, the presence in social networks and, specifically, the inclusion of links to them seems to be more frequently. 100% of the websites analyzed present the possibility of accessing some of these platforms and in addition 78.57% of the hospitals allow users to promote their contents on different social networks. Facebook, with a 96.42 percentage is the social network better positioned on the hospitals’ websites. It is followed by YouTube with 80.35%, Twitter, 64.28% and Instagram with a percentage of 58.92 despite is the more growing social network.

Referring to the type of content is predominantly textual in all cases (100%) and photographic 98.21%. They are followed by audiovisual content with 91% out of the total. Multimedia appears in 76.78% of the websites analyzed. Otherwise, the possibility of commenting on the contents that hospitals publish on their websites is reduced to 32.14% of the total websites studied. The presence of the date of publication does not always accompany the publications, but it is very frequent (it is on 89.28% of the websites); while the use of keywords highlighted in bold, an aspect that promotes reading and positioning in search engines, is also high score with 83.92% in both indicators.

It is worth noting differences arisen from the comparative analysis between the best hospitals in terms of medical services (ranked in “América Economía” portal) and those that present a better digital communication strategy on their websites in our study.

### 4.5. Quality and Accesibility

Regarding the endorsement of some specific quality accreditation seal for the health sector (Hon Code, WMA, ACSA), the study states that only the 62% of the sample have this characteristic. In addition, at the accessibility level, and based on the results obtained from the filtering elaborated by the automatic tool of the Web Accessibility Test (WAT) developed by the Information and Communication Technology Centre (http://www.tawdis.net), all the hospitals analyzed obtain a variable number of problems and warnings in their accessibility component. The evaluation place hospitals, all of them, in a WAI-AA register, which means that they occupy the intermediate position of the TAW evaluation. 

### 4.6. Summarizing Results

According to all data analyzed, a whole of 65 indicators related to six different variables of study, it is expected that the sample studied, which represents the best hospitals in the ranking “America Economía Ranking” have obtained best results in terms of communication and digital strategy from the studied addressed. However, if we compared the results from our study with results published in the ranking, where the best hospitals have been selected for our sample, it can be observed surprising results showing that best hospitals in the ranking in terms of medical services, are not the best hospitals in terms of digital communication strategies. In this regard, as can be seen in Figure 9, the “Albert Einstein Israeli Hospital” in Brazil stands out for occupying the first position in both directories and, therefore, is consolidated as one of the best centers in both dimensions: medical and web communication. Along with this center, only the “Clínica Alemana” in Chile remains among the top positions in both rankings. Specifically, it occupies second place in the list of “América Economía Ranking” and fourth in the one derived from this study. Furthermore, it is very interesting that hospitals such as “Marcelino Chapagnat” in Brazil, which is in 46th place in the “América Economía ranking”, was the second hospital with the best website in all of Latin America. This data shows, therefore, that there is no correlation between the quality of the medical services offered by the hospitals and the quality of their digital communication through their website. This evidence is confirmed by the cases of “Hospital Marcelino Chapagnat” in Brazil, which is ranked second with the best communication on its website, but 46th in the directory prepared with “América Economía”; while in the case of “Hospital General de Medellín Luz Castro Gutiérrez”, which is third with the best website, it is 33th in the ranking on medical services.

## 5. Discussion

Generally speaking, the typology of the contents from analyzed websites need to be reformulated in order to strengthen the human and social dimension in them as highlighted by Casero-Ripollés [6], Arencibia-Jiménez and Aibar-Remón [13] or Gong et al. [22]. In addition to this, there is a nonexistence of commitment to the most innovative contents in the current communication scenario offered by the Internet, such as augmented reality, immersive video or photography, gamification and, especially, the podcast, which has acquired a very wide growth in recent years in other sectors or thematic areas. In line with recent research [2,4,8,33] the role of social networks in all areas of live is increasing, it is striking that Twitter and Instagram are in the latest positions, 41% of hospitals do not have a Instagram account and 36% do not have one in Twitter, aspects that could be linked to the difficulties of health centers to permanently generate interesting content that feeds these networks and could engage with the great amount of users they both have. 

As pointed in previous research from Denecke and Nejdi [10], Wald et al. [14] or Jha et al. [15] hospitals must also be able to make certain content on their websites transparent, such as management organization charts or waiting list data. Similarly, the need is detected in line with Calvo [17] to design studies that analyze the presence and use of organic search engine positioning techniques or SEO, a crucial aspect to increase the reach and visibility on search engines. In an *infodemic* context, as the WHO has pointed out, it is noteworthy that the presence of spaces or content centered on the pandemic (characteristics, advice or recommendations in the event of contagion) and related to COVID-19 disease is not present on all websites. This seem to be an important lacking in a situation in which digital media are the only information ways because of the global lockdown. The seriousness and scope of this virus should lead hospitals to offer service information on the coronavirus placed in their respective websites which would help to provide a better, informed citizenry and also avoid the collapse of many health centers [7,10,21].

Brazil, Chile and Colombia are the countries with the largest number of hospitals in the top positions of the ranking. This aspect allows us to conclude that in the Latin American context, socio-economic differences are projected that are equally identifiable in the digital communication strategies that hospitals apply through their corporate websites. This aspect, considering the postulates of Wald et. al. [14], should lead to a reflection on the importance of digital communication in health centers so that citizens can be informed and make appropriate decisions as pointed by Mariscal [32]. 

In addition, the study allows a diagnostic report to be drawn up on the defining elements of Latin American hospital websites in terms of content and information access, visibility and usability, pointing out their strengths and weaknesses, their most outstanding interactive resources and the type of connection that exists with other platforms, especially social networks. However, there is a need to project this line of research, as they point out [3,15,43], with the aim of delving into the communicative aspects which, especially in the current context, contribute to facing the threat of the *infodemic* which has emerged with the COVID-19 pandemic. In this regard, it is worth noting that COVID-19 outbreak has accelerated the necessity for digital solutions, highlighting the essential role digital information plays in appropriate institutions such as hospitals, in order to face the *infodemic* [3]. Having in mind the global situation of citizens in an unprecedented lockdown, digital media emerge as massive resources of information consumption, so their accuracy and proper development evolve in effective and genuine sources as pointed in recent works published by Hantris et al. [3], Yusof et al. [9], Deneck et al. [10] or Wald et al. [14]. Accordingly, it could be said that communication strategies in hospitals have become fundamental approaches to face the pandemic and avoid misinformation and rumor which, “among other things, hampered public health responses and effective crisis communication by sowing confusion and distrust in official and medical guidance” [44,45]. Even before the pandemic crisis, digital transformation and technological development encompass with communication strategies were essential issues for institutions, nowadays it become fundamental issues not only for citizens information but for counteracting misinformation and *infodemic*.

## 6. Conclusions

Our conclusions point to a demanding situation in which Latin American hospital websites should undergo a substantial review to be restructured and reformulated. In this sense, as pointed out by Calvo-Calvo [17] and Gámez de la Hoz [34], this is an area of study that needs new and renewed research outcomes. In our study it is observed a significant gap between the hospitals that occupy the top positions in the rankings of medical services and those that are placed in the directory of best websites. Results give us a double conclusion: on the one hand, it could be concluded that the hospitals offering better healthcare services have not given digital communication through their websites a leading role. On the other hand, it could be said that the efficiency of the communication strategy is not always linked to economic parameters but derives from other aspects. As pointed out by Gong et al. “[Hospitals] *offer essential medical supports to the public during the COVID-19 outbreak, reduce the social panic, promote social distancing, enhance the public’s ability of self-protection, correct improper medical-seeking behaviors, reduce the chance of nosocomial cross-infection, and facilitate epidemiological screening, thus, playing an important role on preventing and controlling COVID-19*” [22]. 

This study contributes to the area of communication related to healthcare services, describing and giving empirical evidences that could enlighten posterior researches. Nevertheless, it is important noting the limitations of the study, as it is a descriptive-qualitative research it cannot be offered casual links with cross sectional data that could be done in the future. Otherwise, the study presents academic interest providing a needed perspective in which hospital´s websites could be legitime sources for spread healthcare information, which has been evidenced crucial in a lockdown situation with digital media as unique sources for health information consultation. The research offers an open pathway for future researches amplifying the sample and presenting similar studies from other regions.

## Figures and Tables

**Figure 1 ijerph-17-09145-f001:**
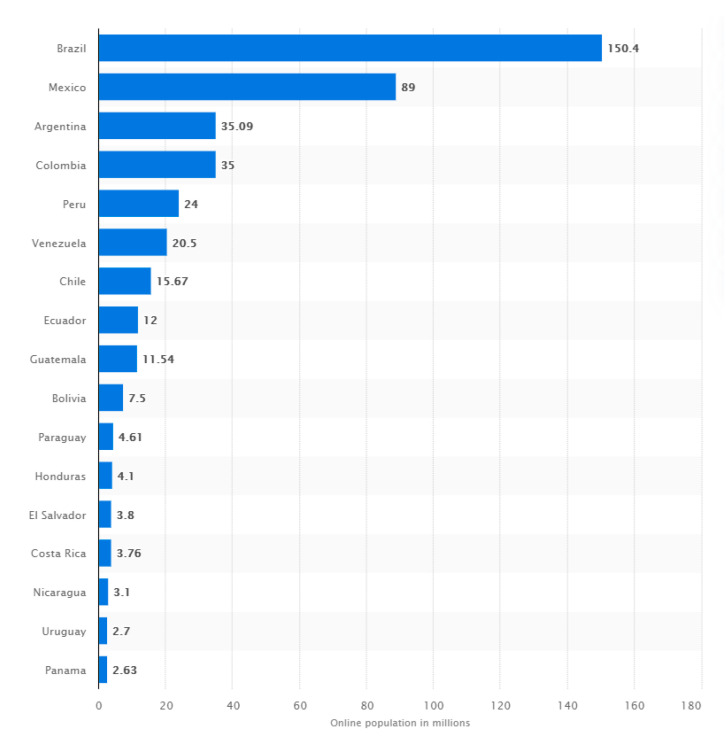
Number of internet users in selected Latin American countries January 2020 (in millions). Source: Statista [38].

**Figure 2 ijerph-17-09145-f002:**
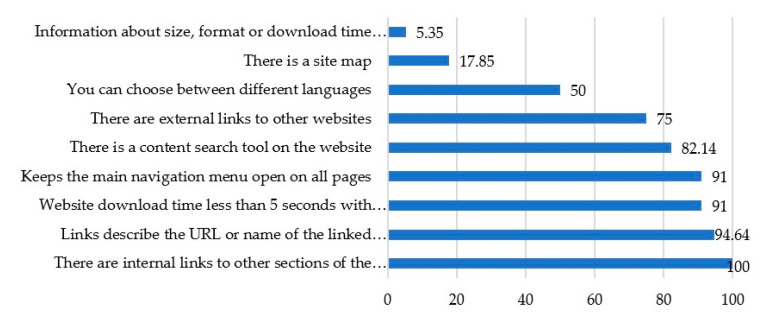
Representing “Usability” scores in percentages for all hospitals from the sample.

**Figure 3 ijerph-17-09145-f003:**
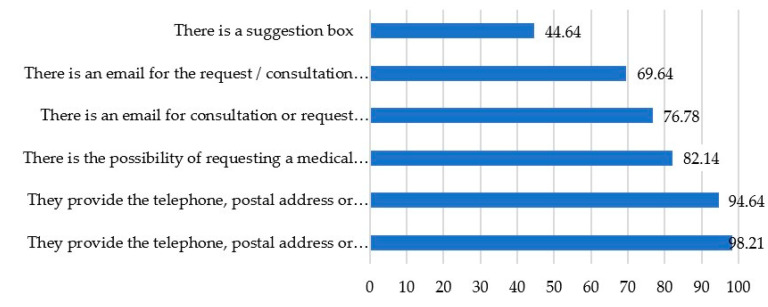
Representation in percentages of “Interactivity and interactions” indicators for all hospitals from the sample.

**Figure 4 ijerph-17-09145-f004:**
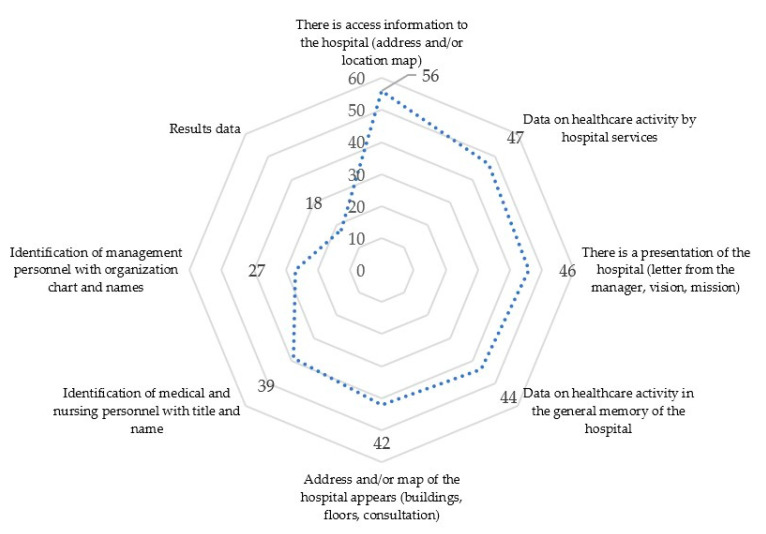
Representation of “Information offered” variable with indicators related in number of hospitals accomplishing them.

**Figure 5 ijerph-17-09145-f005:**
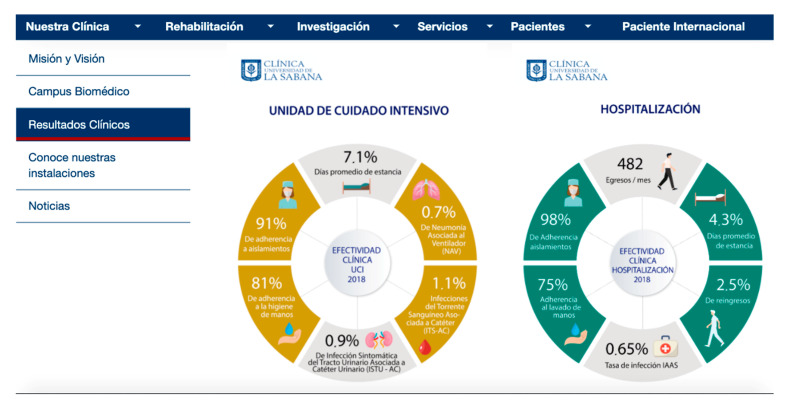
Open and data content of the “Clínica Universidad de la Sabana” (Colombia).

**Figure 6 ijerph-17-09145-f006:**
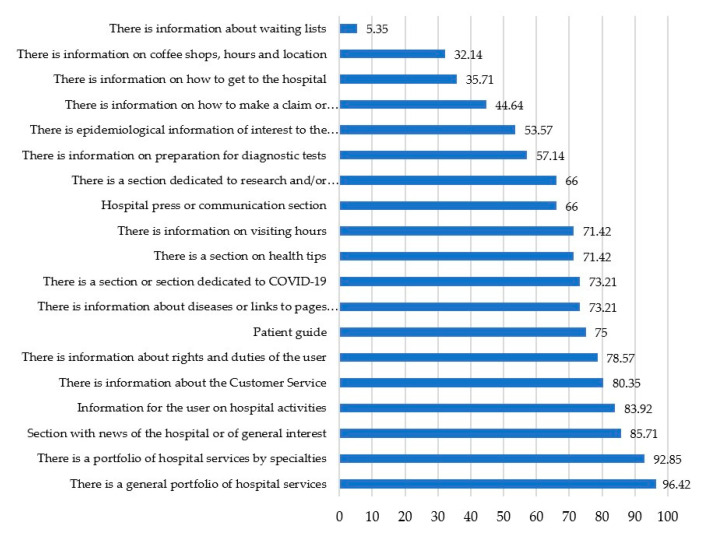
Website information offered. Own elaboration.

**Figure 7 ijerph-17-09145-f007:**
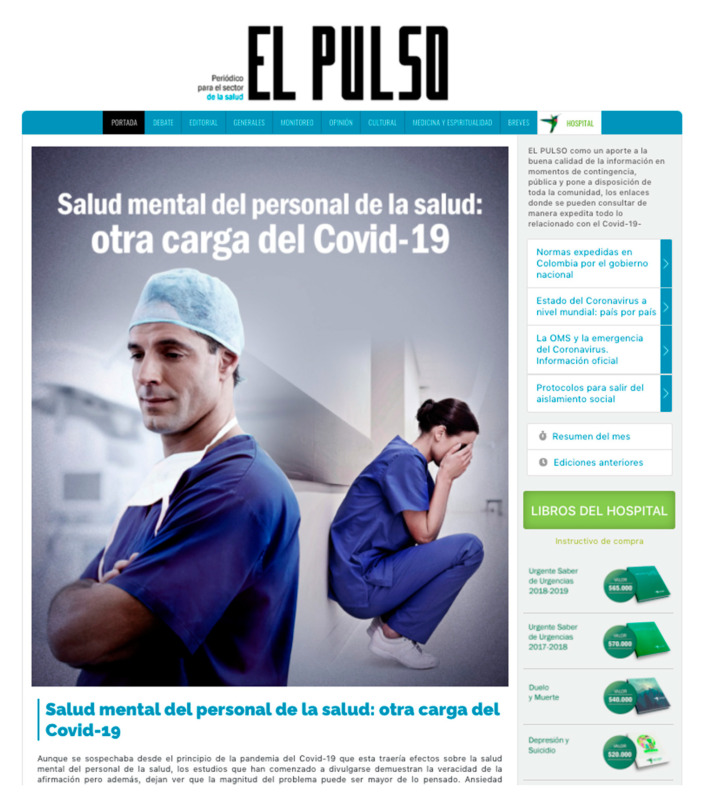
“El Pulso”, online magazine of the Hospital San Vicente Fundación de Medellín (Colombia). Source: Website of the hospital.

**Figure 8 ijerph-17-09145-f008:**
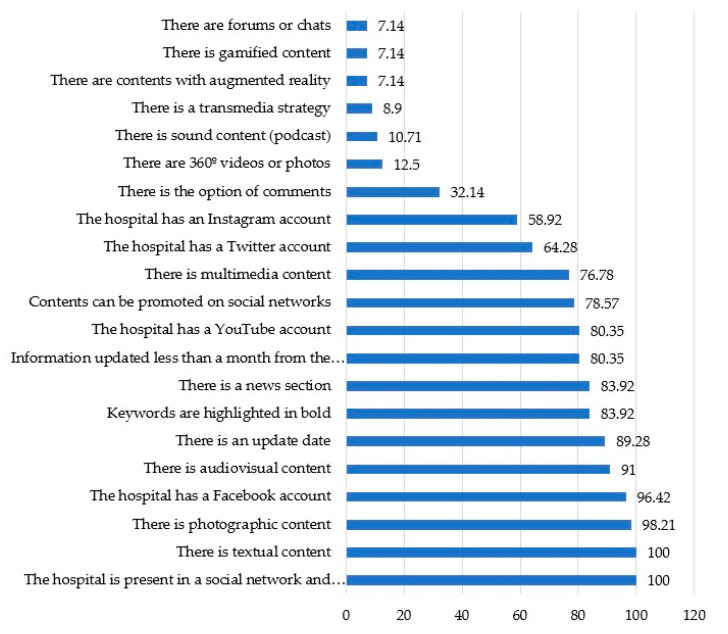
Social media and content updating. Own elaboration.

**Figure 9 ijerph-17-09145-f009:**
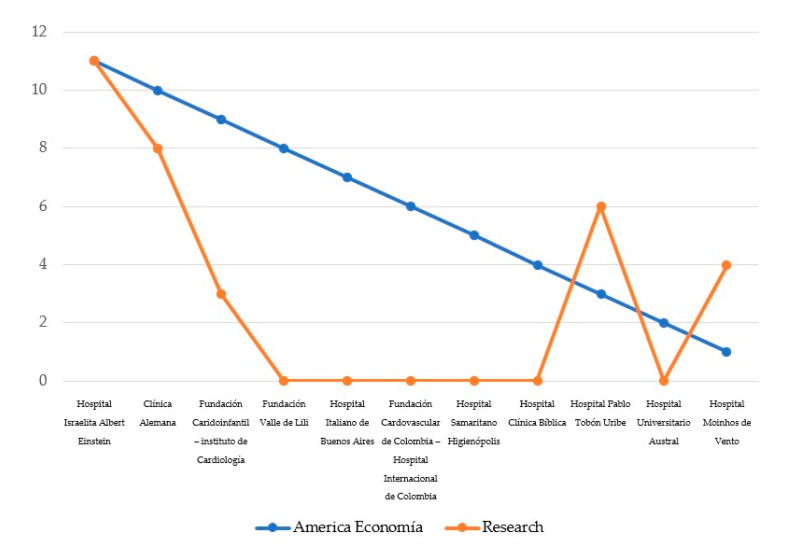
Comparative analysis between the results of the “America Economía Ranking” and results obtained from our study. Own elaboration.

**Table 1 ijerph-17-09145-t001:** Research variables of study and description.

Num.	Research Variable	Variable Description
1	Usability	Intuitive handling and easy navigation through the page
2	Interactivity and relationship with users	Types of virtual exchanges between the hospital’s website and Internet users
3	Information offered and typology	Presence or not of relevant information for the user (about the hospital and about the website)
4	Social media and updating of contents	Types of content published and frequency of publication.
5	Quality references	Voluntary adhesion of the site to some recognized quality seal for health webs
6	Accessibility	Capacity of the website to respond effectively to users with special needs

**Table 2 ijerph-17-09145-t002:** Research variables of study and indicators stablished for each one.

Research Variable	N	Indicators for Each Research Variable	Value
Usability	1	Website download time less than 5 s with ADSL	0/1
	2	Keeps the main navigation menu open on all pages	0/1
	3	There is a content search tool on the website	0/1
	4	They indicate the size, format and/or download time of the file	0/1
	5	There are external links to other websites	0/1
	6	There are internal links to other sections of the hospital’s website	0/1
	7	The links describe the URL or name of the linked website	0/1
	8	There is a site map	0/1
	9	The platform allows you to choose between different languages	0/1
Interactivity and relationship with users	10	There is an email for consultation or request of general information	0/1
	11	They provide the telephone, postal address and/or fax number for general information	0/1
	12	There is e-mail for the request/consultation with the hospital services	0/1
	13	They provide telephone, postal and/or fax information with the hospital services	0/1
	14	There is a suggestion box	0/1
	15	It is possible to make an appointment through the website	0/1
Information offered	16	There is information on how to get to the hospital (address and/or location map)	0/1
	17	Address and/or map of the hospital appears (buildings, floors, office)	0/1
	18	There is a presentation of the hospital (letter from the manager, vision, mission)	0/1
	19	Identification of management personnel with organization chart and names	0/1
	20	Identification of medical and nursing staff with position and name	0/1
	21	Data on care activity by hospital services	0/1
	22	Data on healthcare activity in the hospital’s general report	0/1
	23	Results data from epidemic	0/1
	24	Epidemiological information of interest to the hospital and the population	0/1
	25	There is a section dedicated to the COVID-19	0/1
	26	There is a portfolio of hospital services by specialty	0/1
	27	There is a general service portfolio of the hospital	0/1
	28	There is information on waiting lists	0/1
	29	There is a section on health advice	0/1
	30	Information on preparation for diagnostic tests is available	0/1
	31	There is information about diseases or a link to pages that offer it	0/1
	32	Information for the user about hospital activities	0/1
	33	Section with news from the hospital or of general interest	0/1
	34	Press or communication section of the hospital	0/1
	35	There is a section dedicated to research and/or teaching-training	0/1
	36	Patient guide	0/1
	37	There is information on how to get to the hospital (means of transport)	0/1
	38	There is information on visiting hours	0/1
	39	There is information on the rights and duties of the user	0/1
	40	There is information on the User Support Service	0/1
	41	There is information on cafeterias, opening hours and location	0/1
	42	There is information on how to make a complaint or suggestion	0/1
Typology and updating of contents	43	There is an update date	0/1
	44	Information updated less than one month from the review date	0/1
	45	Keywords are highlighted in bold	0/1
	46	The hospital is present in some social network and there are links to it from the web	0/1
	47	The hospital has a Facebook account	0/1
	48	The hospital has a Twitter account	0/1
	49	The hospital has an Instagram account	0/1
	50	The hospital has a YouTube account	0/1
	51	Textual content exists	0/1
	52	There is photographic content	0/1
	53	Audiovisual content exists	0/1
	54	There is sound content (podcast)	0/1
	55	Multimedia content is available	0/1
	56	There are videos or 360° photographs	0/1
	57	There are contents with Augmented Reality	0/1
	58	There are gamification-based contents	0/1
	59	There is a news section	0/1
	60	There are forums or chats	0/1
	61	There is a transmedia strategy	0/1
	62	Content can be promoted on social networks	0/1
	63	There is the option of comments	0/1
Quality references	64	Website attached to a specific quality accreditation seal for the health sector (Hon Code, WMA, ACSA)	0/1
Accessibility	65	From Web Accessibility Test (TAW)	0/1

Own elaboration from Codina [24,25], Tejedor [42], Calvo-Calvo [17] y Cobos y Recoder [30].

**Table 3 ijerph-17-09145-t003:** Sample from ranking of hospitals in “*América Economía*”.

R	Hospital/Clinical Center	Country	City	Type *	Q. I.
1	Hospital Israelita Albert Einstein	Brasil	Sao Paulo	Private	98.45
2	Clínica Alemana	Chile	Santiago	Private	90.21
3	Fundación Caridoinfantil–instituto de Cardiología	Colombia	Bogotá	U. Private	83.61
4	Fundación Valle de Lili	Colombia	Calí	U. Private	83.60
5	Hospital Italiano de Buenos Aires	Argentina	Buenos Aires	Private	80.86
6	Fund. Cardovascular de Colombia–Hospital Internacional de Colombia	Colombia	Bucaramanga	U. Private	77.78
7	Hospital Samaritano Higienópolis	Brasil	Sao Paulo	Private	77.70
8	Hospital Clínica Bíblica	Costa Rica	San José	Private	76.98
9	Hospital Pablo Tobón Uribe	Colombia	Medellín	U. Private	76.63
10	Hospital Universitario Austral	Argentina	Buenos Aires	Private	76.39
11	Hospital Moinhos de Vento	Brasil	Porto Alegre	Private	76.38
12	Centro Médico Imbanaco	Colombia	Calí	Private	76.38
13	Médica Sur	México	Ciudad México	Private	74.86
14	Hospital Alemâo Oswaldo Cruz	Brasil	Sao Paulo	Private	74.54
15	Clínica Ricardo Palma	Perú	Lima	Private	74.28
16	Hospital Universitario San Vicente Fundación	Colombia	Medellín	U. Private	73.22
17	Clínica Internacional	Perú	Lima	Private	73.16
18	Hospital Metropolitano	Ecuador	Quito	Private	72.83
19	Pacífica Salud Hospital Punta Pacífica	Panamá	C. de Panamá	Private	72.75
20	Clínica San Pablo	Perú	Lima	Private	72.51
21	Hospital Infantil Sabara	Brasil	Sao Paulo	Private	70.73
22	Hospital Sâo Vicente de Paulo	Brasil	Río Janeiro	Private	67.27
23	Hospital Santa Paula	Brasil	Sao Paulo	Private	66.70
24	Complejo Asistencial Dr. Sótero del Río	Chile	Santiago	Private	65.97
25	Ciudad Universidad de la Sabana	Colombia	Bogotá	U. Private	65.31
26	Clínica Las Américas	Colombia	Medellín	Private	65.30
27	Hospital Municipal Dr. Moysés Deutsch—M’Boi Mirim	Brasil	Sao Paulo	Public	64.93
28	Clínica Universitaria Bolivariana	Colombia	Medellín	U. Private	64.80
29	Hospital Zambrano Hellion–Tec Salud	México	San Pedro Garza	U. Private	64.71
30	Hospital Edmundo Vasconcelos	Brasil	Sau Paulo	Private	64.64
31	Clínica del Occidente	Colombia	Bogotá	Private	64.56
32	Méderi	Colombia	Bogota	U. Private	64.47
33	Hospital General de Medellín Luz Castro de Gutiérrez	Colombia	Medellín	Public	64.09
34	Hospital Infantil Teletón de Oncología	México	Querétaro	Private	63.69
35	Hospital El Cruce Dr. Nestor Carlos Kirchner	Argentina	Florencio Varela	U. Public	63.68
36	Hospital San José Tec Salud	México	Monterrey	U. Private	63.35
37	Hospital 9 de Julho	Brasil	Sao Paulo	Private	63.20
38	Sanatorio La Costa – Grupo San Roque	Paraguay	Asunción	Private	62.27
39	SES Hospital de Caldas	Colombia	Manizales	Public	61.48
40	Hospital Galenia	México	Cancún	Private	61.35
41	Clínica El Rosario Sede Tesoro	Colombia	Medellín	Private	61.03
42	Centro Cardiovascular Colombiano Clínica S. María	Colombia	Medellín	Private	60.87
43	Clínica Medellín	Colombia	Medellín	Private	60.50
44	Hospital Samaritano Paulista	Brasil	Sao Paulo	Private	60.21
45	Fundación Hospital Infantil Los Ángeles	Colombia	Pasto	Private	59.74
46	Hospital Marcelino Champagnat	Brasil	Curritiba	Private	59.45
47	Hospital Brasilia	Brasil	Brasilia	Private	59.07
48	Hospital Universitario Departamental de Nariño	Colombia	Pasto	U. Public	58.47
49	Hospital Universitario Infantil de San José	Colombia	Bogotá	U. Private	58.09
50	Clínica de Marly	Colombia	Bogotá	Private	57.73
51	Clínica Los Nogales	Colombia	Bogotá	Private	57.34
52	Hospital Samaritano Botafogo	Brasil	Sao Paulo	Private	56.87
53	Clínica Las Vegas	Colombia	Medellín	Private	56.49
54	Sanatorio Güemes	Argentina	Buenos Aires	Private	56.20
55	Hospital Universitario Clínica San Rafael	Colombia	Bogotá	Private	54.99
56	Hospital Sao Lucas Copacabana	Brasil	Río de Janeiro	Private	54.92
57	Centro Policlínico del Olaya	Colombia	Bogotá	Private	53.34
58	Hospital Porto Días	Brasil	Belem	Private	51.11

* Type: U. Private = Private University Hospital.

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
