# Peer review of "Tracking Websites’ Digital Communication Strategies in Latin American Hospitals During the COVID-19 Pandemic"

_ijerph, 2020, doi:10.3390/ijerph17239145_

Round 1

Reviewer 1 Report

The manuscript received for review addresses essential aspects of the best hospitals' website communication strategies in Latin America. The work is original, interestingly constructed. Apart from minor comments as below, there is no need to be very critical of the content. Nevertheless, in order to publish it, it requires serious editing in terms of the English language. Some parts of it are hard to understand. Often the authors use an excess of words- the paragraphs are too wordy. Sometimes they use words not used in a given context. There are many grammatical errors. The Authors generally use too many long, complex sentences.
Detailed minor notes:
1 / line 127-128- the name of the country in the 2nd position is missing
2 / Unnecessary statements under the tables: Own elaboration
3 / Figures are missing scale -%, points, other.
4 / table 2- the expression "gamma contents" is not understood - needs some explanation
5 / table 3- it is not clear what is meant by U. private, U.public.

Author Response

Dear reviewer, thanks a lot for your suggestions that improve the quality of our manuscript. We have striven to accomplish all your valuable suggestions. We detail as follows the changes implemented in the paper:

Point 1: English editing

Extensive English editing has been applied as can be seen in track changes document. We have shortened sentences in order to make paragraph more understandable and replace some repeated words. Some paragraphs have been rewritten as you can check in the track-changes document.

Point 2: Detailed minor notes:

1 / line 127-128- the name of the country in the 2nd position is missing

It has been added “Mexico” as second country missing

2 / Unnecessary statements under the tables: Own elaboration

Sure, it has been removed from all tables and figures

3 / Figures are missing scale -%, points, other.

It has been added to Figures footnote the scales (percentages)

4 / table 2- the expression "gamma contents" is not understood - needs some explanation

The expression has been clarified: gamification-based content

5 / table 3- it is not clear what is meant by U. private, U.public.

It has been added as table footnote note *U.Private = University Private Hospital

Reviewer 2 Report

This is a compelling study that will certainly attract a high level of interest because of its topic. But its most significant finding, that the best hospitals in Latin America (with one exception) have the least developed websites, needs to be explored in its full complexity. The most interesting possible conclusion here, which you do briefly touch upon, seems to me not that digital communications strategies "must" be improved across the board, but rather that these innovations and media may serve as a surrogate for quality healthcare, or even as a kind of PR and propaganda for hospitals with a lower ranking. We need greater presentation and reflection of evidence that digital strategies--including exotic new possibilities such as AR and "gamification"--really improve healthcare. 

Another point that needs to be considered: why now? Do the sanitary considerations of COVID-19 truly warrant this transition? Or do new technological conditions simply make it possible? Could the digitization not only of hospital information, but also basic health practices actually dissuade people from getting the care they need (by acting as a surrogate)? One worries that hospitals getting more involved in digital communication may actually contribute to the "infodemic" decried in the paper and by the WHO, especially if/when shared out of context on social media.

There is considerable potential here, as long as these findings are explored in the broader context of technical communication. One text you should explore, which I cannot recommend more avidly, is Steven Katz's classic article on tech comm "The Ethic of Expediency." One worries that more expedient digital health communication may in fact proceed at the peril Katz describes.

Thank you for the opportunity to read this work. Please note that I have also made significant in-text suggestions on the manuscript itself. 

Author Response

Dear reviewer, thanks a lot for your suggestions that improve the quality of our manuscript. We have striven to accomplish all your valuable suggestions. We detail as follows the changes implemented in the paper:

Point 1: English editing

Response 1: Extensive English editing has been applied as can be seen in track changes document. We have shortened sentences in order to make paragraph more understandable and replace some repeated words. Some paragraphs have been rewritten as you can check in the track-changes document.

Point 2: This is a compelling study that will certainly attract a high level of interest because of its topic. But its most significant finding, that the best hospitals in Latin America (with one exception) have the least developed websites, needs to be explored in its full complexity. The most interesting possible conclusion here, which you do briefly touch upon, seems to me not that digital communications strategies "must" be improved across the board, but rather that these innovations and media may serve as a surrogate for quality healthcare, or even as a kind of PR and propaganda for hospitals with a lower ranking. We need greater presentation and reflection of evidence that digital strategies--including exotic new possibilities such as AR and "gamification"--really improve healthcare. 

Response 2: We really appreciate your considerations on the interest of the study, your comments made us add some interesting reflections on how digital communications strategies could improve the healthcare attention for patients and the healthcare system:

Lines 106-112 “These reflections are supported by previous literature as shown in the extensive literature review published by Kampmeijer et al. [21] in which authors evidenced how e-health and m-health tools (understood as namely apps, websites, digital content, video consults and webinars) truly improve healthcare quality for adults in the way they receive more supportive assistance and are better informed. Moreover, since COVID-19 lockdown situation occurred the role of institutional websites from hospitals, has become an essential issue, in most cases the unique communication channel between patients and healthcare services [22, 23]”.

Lines 122-131: “Digital media and connected devices have evolved to a ubiquitous part of our daily life and the Internet have paved the way to a wider interactive communication in the relation with citizens [2]. Healthcare services delivery and information availability supported by digital technologies such as hospital´s websites are key issues not only in the enhancement of patient satisfaction but in the efficiency of e-Health systems [20-22]. As stated from Kotsiliers et al. [33] if these means are properly designed, they offer a double innovative pathway: from one hand giving patients the power not only to educate themselves but take part in the decision-making process of their health; and in the other hand these new channels for communication and interaction make healthcare stakeholders gain in essential aspects such as cutting costs, gaining efficiency and improving the healthcare system”.

Point 3: Another point that needs to be considered: why now? Do the sanitary considerations of COVID-19 truly warrant this transition? Or do new technological conditions simply make it possible? Could the digitization not only of hospital information, but also basic health practices actually dissuade people from getting the care they need (by acting as a surrogate)? One worries that hospitals getting more involved in digital communication may actually contribute to the "infodemic" decried in the paper and by the WHO, especially if/when shared out of context on social media.

Response 3: We consider the accuracy of this research now is justified, as expressed in the introduction, because “COVID-19 has had a global impact on different social areas of our daily lives, such as mobility, leisure, entertainment, education [1], communication [2], economics, market, and, especially, health sector. Hospitals have been overwhelmed by the emergence of cases by coronavirus disease. Access to clear, structured and accurate information has become a decisive resource for citizens and a crucial challenge for health centers to ensure some kind of social order in a situation of lockdown. In this regarding sanitary centers such as hospitals accomplish digital strategies that really allow citizens to be properly informed they are directly fighting the infodemic. Hospitals are main sources authenticated to give sanitary information

Point 4: There is considerable potential here, as long as these findings are explored in the broader context of technical communication. One text you should explore, which I cannot recommend more avidly, is Steven Katz's classic article on tech comm "The Ethic of Expediency." One worries that more expedient digital health communication may in fact proceed at the peril Katz describes.

Response 4: Although Steven Katz's in his “The Ethic of Expediency” on the rhetoric of technical communication during the Holocaust has become a reference point for discussions of ethics, we would light to highlight the differences among discourses. So, the unethical issues of the digital communications from hospitals in their web sites are out of this point or at least this is not the focus of the research presented, which is undeniably very interesting observation.

Point 5: Please, note that we have make changes attending your suggestions in pdf document:

Response 5: Line 17, 18 in the Abstract

Line 36, eloquent changed by accurate

Line 66, “due the enormous strain for COVID-19 patients”

Line 76, is clarified “opens up an aspicious field of study focusing on the relevance of this sites to fight against the infodemic”

Line 99, the paragraph has been re-written

Line 146-147: following your suggestion we have added “…pointed out that 78% of young –less affected population by COVID-19– Latin Americans between…”

Line 166: we certainly have removed the article from “the COVID-19”

Line 154, regarding to this comment, it was certainly unclear. Now “The study analyzes the appearance or not of COVID-19, taking into consideration the presence of information related to the pandemic on the websites analyzed

Line 181, Table 2, Point 10 “an email”, Pint 23 “Results data from epidemic”

Line 230, we have replaced the figure showing all data in order to avoid misleading as you kindly suggested.

Line 261: the concept of infodemic that operates in this case is the one defined by WHO  “alluding to the rapid growth of manipulated and falsified content circulating, especially, in cyberspace”

Line 268, regarding this comment we have clarified the sentence: “Having in mind the situation has provoked many collapses in several hospitals, the information, regarding to pandemic (prevention, actions, tests, disease characteristics, etc.) becomes essential and invites a deep reflection from the communication perspective”

Line 269: regarding this comment we certainly agree with you, but our objective in this point is just describe not judge if this is just a king of PR, a way of showing how sleek and modern this hospital is, undoubtedly is.

Line 277, Are you saying that the presence of such contents would be a good thing? If so, why? In this results section we are not pretending to express opinions on the results from analysis, we pretend to be objective and descriptive in this sentence: “In general, less than a dozen of the hospitals analyzed present some content of this type. In addition, chats and forums have lost their importance as spaces for dialogue and only appear in four of the centers analyzed”.

Line 299, the transmedia strategy indicator has been clarified “which is related to information offered in different digital platforms”.

Line 325: It is important noting that our research does not state that hospitals would improve their healthcare service from the developing of their websites or that these pages will be a surrogate for quality of healthcare. The point is that in a global pandemic situation where the most part of population in the world lived, and still live, an unprecedent lockdown, the hospital´s websites, which are epicenter of the pandemic in many aspects, may play an important rol due people just can communicate or be informed through digital media.

Line 360, Conclusions, regarding the first and second comment it has been added: “As pointed by Gong et al. “offer essential medical supports to the public during the COVID-19 outbreak, reduce the social panic, promote social distancing, enhance the public’s ability of self-protection, correct improper medical-seeking behaviors, reduce the chance of nosocomial cross-infection, and facilitate epidemiological screening, thus, playing an important role on preventing and controlling COVID-19” [22]”. And Line 366 has been completed.

Line 353, plural deleted “research”

Line 360, “it is striking” replaced by “it is noteworthy” more suitable expression.

Line 379, it has been explained in more detail as suggested: “Having in mind the global situation of citizens in an unprecedented lockdown, digital media emerge as massive resources of information consumption, so their accuracy and proper development evolve in effective and genuine sources as pointed in recent works published by Hantris et al. [3], Yusof et al. [9], Deneck et al. [10] or Wald et al. [14].

Please see the attachment with all track changes

Reviewer 3 Report

Authors should add limitations of the study section in discussion and conclusions.

Comments on “ Tracking websites digital communication strategies in Latin American hospitals during COVID-19.

General comments: The manuscript addresses the importance of communication strategies from hospitals in the COVID19 period in South America. The work looks original and might prove to be interesting for readers. However, study method is not stated in the abstract. This is a qualitative study with descriptive components. Study design should be clarified. The size of the article can be reduced further, by extensive editing.

Specific comments:

Abstract: Authors should divide abstract into introduction, method, results and conclusions. Study design method is not seen. Authors should name the study design for example qualitative or quantitative.

Introduction: Research questions were stated at the end. Authors should discuss the importance of those questions for the study and their relation to the variables chosen for the study.

Study method: Authors should provide specific inclusion and exclusion criteria for the study sample or indicators chosen for the study. Authors should use either variables or indicators word consistently.

Strengths and Limitations: please provide strengths and limitations section at the end of discussion.

Discussion: Authors should divide discussion and conclusions section separately.

The picture above line 270 looks unnecessary.

Author Response

Dear reviewer, thanks a lot for your suggestions that improve the quality of our manuscript. We have striven to accomplish all your valuable suggestions. We detail as follows the changes implemented in the paper:

English editing

Extensive English editing has been applied as can be seen in track changes document. We have shortened sentences in order to make paragraph more understandable and replace some repeated words. Some paragraphs have been rewritten as you can check in the track-changes document.

General comments:

Point 1: The manuscript addresses the importance of communication strategies from hospitals in the COVID19 period in South America. The work looks original and might prove to be interesting for readers. However, study method is not stated in the abstract. This is a qualitative study with descriptive components. Study design should be clarified. The size of the article can be reduced further, by extensive editing.

Response 1: Abstract has been improved following your suggestions. The study designed, as expressed in the abstract and then in the Methodology paragraph was based in previous research focused on analyzing websites.

Specific comments:

Point 1: Abstract: Authors should divide abstract into introduction, method, results and conclusions. Study design method is not seen. Authors should name the study design for example qualitative or quantitative.

Response 1: Abstract has been improved following your suggestions, firstly, describing the qualitative methodology and, secondly, adding the conclusions from the study that were lacking.

Point 2: Introduction: Research questions were stated at the end. Authors should discuss the importance of those questions for the study and their relation to the variables chosen for the study.

Response 2: Following your suggestions we really agree and have deleted research questions from Introduction. Instead, we completed the introduction, as indicated in Authors guidelines from IJERPH and added some reflections from conclusions in order to make introduction more understandable. You can check it lines 77-81

Point 3: Study method: Authors should provide specific inclusion and exclusion criteria for the study sample or indicators chosen for the study. Authors should use either variables or indicators word consistently.

Response 3: In this regard the approaches to the research has been described and defined using a step-by-step methodology from reviewing specific literature, adapting an existing model, the SAAMD pattern, and finally, defining and designing adapted indicators (Table 2) previously defined six variables of study. All possible criteria of study have been included, and researchers had decided the sample of study in a convenience way. Future studies will focus on other geographical areas. This is one of the important limitations in the study, sample could be wider. But it is a beginning.

Point 4: Strengths and Limitations: please provide strengths and limitations section at the end of discussion. Discussion: Authors should divide discussion and conclusions section separately.

Response 4: As suggested we have divided Discussion (line 359) and Conclusion (line 407) sections and added content related to strengths and limitations of the research presented. Lines 422-430

Point 5: The picture above line 270 looks unnecessary.

Response 5: Maybe this Figure could seem unnecessary but we really consider it illustrates how a hospital can have its own resources for communication with patients. So, if you do not mind, we prefer to maintain it.

Reviewer 4 Report

The present study shows a well developed and clear academic structure.
The topic presented is up to date and interesting, with a review of current
and abundant literature. This allows the reader to see the relevance of
the subject over the years. The argumentative ability is demostrated
although some aspects related to English could be improved.
The methodology is well developed and argued. Both the instruments and
the sample are justified. It is recommended to review the conclusions,
as they are brief for the interesting results obtained.

Author Response

Dear reviewer, thanks a lot for your suggestions that improve the quality of our manuscript. We have striven to accomplish all your valuable suggestions. We detail as follows the changes implemented in the paper: 

Point 1: English editing

Response 1: Extensive English editing has been applied as can be seen in track changes document. We have shortened sentences in order to make paragraph more understandable and replace some repeated words. Some paragraphs have been rewritten as you can check in the track-changes document.

Point 2: It is recommended to review the conclusions, as they are brief for the interesting results obtained. 

Response 2: The Conclusions has been amplified and corrected in different points as follow. It is worth noting that a Discussion paragraph has been added separately from the Conclusion paragraph in order to make more sense:

Line 360, it has been added following suggestion: “As pointed by Gong et al. “offer essential medical supports to the public during the COVID-19 outbreak, reduce the social panic, promote social distancing, enhance the public’s ability of self-protection, correct improper medical-seeking behaviors, reduce the chance of nosocomial cross-infection, and facilitate epidemiological screening, thus, playing an important role on preventing and controlling COVID-19” [22]”. And Line 366 has been completed.

Line 353, plural deleted “research”

Line 360, “it is striking” replaced by “it is noteworthy” more suitable expression.

Line 379, it has been explained in more detail as suggested: “Having in mind the global situation of citizens in an unprecedent lockdown, digital media emerge as massive resources of information consumption, so their accuracy and proper development evolve in effective and genuine sources as pointed in recent works published by Hantris et al. [3], Yusof et al. [9], Deneck et al. [10] or Wald et al. [14].

Line 410: new paragraph added separately from Discussion

It has been added also a paragraph describing strengths and limitations of the study:

Line 425-432: This study contributes to the area of communication related to healthcare services, describing and giving empirical evidences that could enlighten succeeding researches. Nevertheless, it is important noting the limitations of the study, as it is a descriptive-qualitative research we cannot offer casual links with cross sectional data that could be done in the future. Otherwise, the study presents academic interest providing a needed perspective in which hospitals are legitime sources for spread healthcare information, which has been evidenced crucial in a lockdown situation with digital media as only sources for information consultation. The research offers an open pathway for future researches amplifying the sample and presenting similar studies from other regions.

Round 2

Reviewer 2 Report

Thank you for your response, and I am glad these suggestions were helpful. Your revisions have addressed many of the concerns I raised, although my main point remains more or less unaddressed: that hospital communications don't exist in a vacuum, nor as a one-directional vector running from an information source to patients/consumers. Rather, they respond to, participate in, and can constitute broader public health conditions and trends (for better or for worse). You claim that you don't intend to "judge" the role of these communications, but the very approach you suggest here implies a tacit approval. 

I am fine with the article being published in its revised form, but would encourage the authors to reconsider whether an argument like that offered by Katz really is outside the present "discourse." In fact, the peril of pushing aside the broader impact and ethical context of too-expedient communications--which surely would include digital communications from hospitals of the sort described in this piece--is exactly Katz's point. 

The authors should consider the possibility that "improved" digital healthcare communications could stand in as a surrogate for actually improved healthcare, which might even exacerbate the COVID situation. Advanced digital communications phenomena, especially when they begin to incorporate features such as AR and telemedicine, may not only promote "some kind of social order in a situation of lockdown"; they become a part of what makes lockdown possible for a virus with the profile of COVID-19. Indeed, the way "lockdown" is discussed in this article and in the authors' response is a little alarming. Lockdown isn't just a neutral "situation," like a natural background parameter. It's a policy, inflected by human interests, and facilitated by what technology makes possible. (i.e., if there was no lockdown for previous pandemics, that's in part because there was no Zoom, no telemedicine, no technological surrogates for essential services including healthcare--and certainly no elaborate hospital websites of the sort tacitly advocated by this study.) 

But again, I am fine with the work being published in its revised form, as long as the authors recognize and reflect upon these limitations. Thank you for the opportunity to read and to respond to this work.